# *EDA* Variants Are Responsible for Approximately 90% of Deciduous Tooth Agenesis

**DOI:** 10.3390/ijms251910451

**Published:** 2024-09-27

**Authors:** Lanxin Su, Bichen Lin, Miao Yu, Yang Liu, Shichen Sun, Hailan Feng, Haochen Liu, Dong Han

**Affiliations:** 1Department of Prosthodontics, Peking University School and Hospital of Stomatology & National Center for Stomatology & National Clinical Research Center for Oral Diseases & National Engineering Research Center of Oral Biomaterials and Digital Medical Devices & Central Laboratory, Beijing 100081, China; slancy1219@pku.edu.cn (L.S.); yumiao@bjmu.edu.cn (M.Y.); pkussliuyang@bjmu.edu.cn (Y.L.); sun_shichen@126.com (S.S.); kqfenghl@bjum.edu.cn (H.F.); 2First Clinical Division, Peking University School and Hospital of Stomatology & National Center for Stomatology & National Clinical Research Center for Oral Diseases & National Engineering Research Center of Oral Biomaterials and Digital Medical Devices & Central Laboratory, Beijing 100081, China; stephanlin@bjmu.edu.cn

**Keywords:** anodontia, deciduous tooth agenesis, *EDA*, genotype–phenotype analysis, hypodontia, oligodontia

## Abstract

Deciduous tooth agenesis is a severe craniofacial developmental defect because it affects masticatory function from infancy and may result in delayed growth and development. Here, we aimed to identify the crucial pathogenic genes and clinical features of patients with deciduous tooth agenesis. We recruited 84 patients with severe deciduous tooth agenesis. Whole-exome and Sanger sequencing were used to identify the causative variants. Phenotype–genotype correlation analysis was conducted. We identified 54 different variants in 8 genes in 84 patients, including *EDA* (73, 86.9%), *PAX9* (2, 2.4%), *LRP6* (2, 2.4%), *MSX1* (2, 2.4%), *BMP4* (1, 1.2%), *WNT10A* (1, 1.2%), *PITX2* (1, 1.2%), and *EDARADD* (1, 1.2%). Variants in *ectodysplasin A* (*EDA*) accounted for 86.9% of patients with deciduous tooth agenesis. Patients with the *EDA* variants had an average of 15.4 missing deciduous teeth. Mandibular deciduous central incisors had the highest missing rate (100%), followed by maxillary deciduous lateral incisors (98.8%) and mandibular deciduous lateral incisors (97.7%). Our results indicated that *EDA* gene variants are major pathogenic factors for deciduous tooth agenesis, and *EDA* is specifically required for deciduous tooth development. The results provide guidance for clinical diagnosis and genetic counseling of deciduous tooth agenesis.

## 1. Introduction

Deciduous tooth agenesis is a severe craniofacial developmental defect, even worse than that of permanent teeth. Due to this disease, physiological functions such as mastication and pronunciation are affected from childhood, and the physical development of patients is delayed. Abnormalities in facial appearance can also have an impact on the social lives of patients, giving rise to psychological disorders [1,2]. Moreover, patients with severe deciduous tooth agenesis are often associated with systemic symptoms, as well as more serious permanent tooth agenesis, resulting in severe alveolar bone atrophy and maxillofacial bone underdevelopment [3]. Furthermore, the necessity for long-term multidisciplinary treatment from an early age creates a significant economic burden on the family. Consequently, deciduous tooth agenesis represents a severe oral disease and hereditary developmental defect that poses a significant risk to human health.

Previous studies on tooth agenesis have concentrated on permanent teeth. The prevalence and classification of tooth agenesis have been based on numerous studies of permanent teeth. According to the number of missing permanent teeth (excluding the third molars), tooth agenesis can be classified as hypodontia (fewer than six missing permanent teeth), oligodontia (six or more missing permanent teeth), and anodontia (missing all the permanent teeth) [4]. However, no classification defines the severity of the congenital absence of deciduous teeth. Based on previous studies, the prevalence of hypodontia ranges from 2.1 to 10.1% in different geographic regions and ethnicities [5,6,7], and that of oligodontia is estimated to be only 0.08 to 0.25% [8,9,10,11,12]. Incidentally, the reported prevalence of the congenital absence of deciduous teeth ranges from 0.4 to 2.4% [13,14,15].

Dental development is a complex process regulated by multiple genes. Over 300 genes contribute to tooth development, and 20 different genes have been confirmed to be associated with permanent tooth agenesis [3,16]. Several studies have shown that *WNT10A*, *PAX9*, *MSX1*, *AXIN2*, and *EDA* are the main causative genes of oligodontia [3,17,18]. The human *ectodysplasin A* (*EDA*) gene is located on chromosome Xq12-q13.1 and encodes proteins that belong to the tumor necrosis factor (TNF) family [19,20]. EDA is a 391 amino acid residue-long type II membrane protein that contains four major domains: transmembrane (TM), furin cleavage, collagen, and TNF homologous domains [21]. TM is associated with transmembrane transport [21,22]. The furin cleavage domain is the recognition site for the cleavage of EDA into soluble secreted protein, which is necessary for EDA–ectodysplasin A receptor (EDAR) binding [22]. The collagen domain promotes the multimerization of the TNF homologous domain for its proper function, and the TNF homologous domain forms a homotrimer that can bind to EDAR [21,22,23]. These domains are crucial for the proper formation and function of teeth and are hot spots for tooth agenesis-causing variants [22,23]. EDA binds to EDAR and subsequently activates the NF-κB signaling pathway [24,25].

A previous study suggests different pathogenic mechanisms between permanent and deciduous tooth agenesis [26]. However, only a few case reports indicate that deciduous tooth agenesis may be caused by *EDA*, *EDAR*, *EDARADD*, *WNT10A*, *PAX9*, *KREMEN1*, and *PITX2* variants [27,28,29,30,31]. To date, no comprehensive studies of the genes responsible for deciduous tooth agenesis have been conducted, and the connection or difference in genetic regulation between deciduous and permanent tooth agenesis has not been studied in depth.

Therefore, to identify the crucial pathogenic genes and clinical features of patients with deciduous tooth agenesis, this study investigated the gene variants in a large cohort of 84 patients with deciduous tooth agenesis.

## 2. Results

### 2.1. Variants in Patients with Deciduous Tooth Agenesis

A total of 54 different variants of 8 genes were identified in 83 patients, including *EDA* (73, 86.9%), *PAX9* (2, 2.4%), *LRP6* (2, 2.4%), *MSX1* (2, 2.4%), *BMP4* (1, 1.2%), *WNT10A* (1, 1.2%), *PITX2* (1, 1.2%), and *EDARADD* (1, 1.2%), whereas 1 patient (1, 1.2%) did not harbor any of the variants (Figure 1A,B). Among the 54 variants we identified, 9 *EDA*, 1 *PAX9*, 1 *LRP6*, 1 *MSX1*, and 1 *EDARADD* variant were novel (Table 1). Details regarding the dental phenotype of the 84 patients in our study are presented in Appendix A [26,30,32,33,34,35,36,37,38,39,40,41,42]. A total of 68 (81.0%) patients with deciduous tooth agenesis were associated with systemic symptoms, and only 16 (19.0%) patients simply presented the congenital absence of teeth (Appendix A).

### 2.2. Clinical Findings in Patients with EDA Variants

The 73 patients with *EDA* variants were all males, aged 2–25 years old. Among these patients, 63 were diagnosed with X-linked hypohidrotic ectodermal dysplasia (XLHED; MIM #305100), and 10 were diagnosed with non-syndromic tooth agenesis (NSTA). Furthermore, 71.2% (52/73) of the patients had a tooth agenesis family history or had mothers with the same variation (Table 1). In patients with XLHED, ectodermal abnormalities were observed, including hair thinning in 50 patients, sparse eyebrows in 22 patients, hypohidrosis in 30 patients, anhidrosis in 21 patients, xerostomia in 19 patients, dry eyes in 13 patients, and xeroderma in 12 patients. A few patients exhibited palmoplantar keratosis, facial erythema, perioral or periocular hyperpigmentation, eczema, nasal collapse, and thin nails (Appendix A).

The patients in this study had an average of 14.1 missing deciduous teeth and 21.7 missing permanent teeth. Of these, the patients with *EDA* variants had 15.4 (4 to 20) missing deciduous teeth. Patients with NSTA had an average of 11.8 missing deciduous teeth, whereas those with XLHED had an average of 15.9 (*p* > 0.05). Meanwhile, 73.0% (73/100) of the remaining anterior deciduous teeth were cone-shaped (Table 1). For permanent teeth, the patients with *EDA* variants had an average of 22.2 (12 to 28) missing teeth, with averages of 17.1 and 23.1 missing teeth in patients with NSTA and XLHED, respectively.

### 2.3. EDA Variants Analysis

Variants in the *EDA* gene were present in 86.9% of patients with deciduous tooth agenesis. Among these *EDA* variants, 29 were missense (67.4%), 8 were frameshift (18.6%), 5 were nonsense (11.6%), and 1 was a splicing variant (2.3%) in intron 2 (Figure 1C). Compound hemizygous variants [c.656C > A (p.Pro219His) and c.661G > C (p.Gly221Arg)] in patient #37 and [c.673C > T (p.Pro225Ser) and c.676C > T (p.Gln226*)] in patient #42 were detected (Table 1). Variants were identified in all 8 exons of *EDA*, with 10 located in exon 4 (23.3%) and 9 located in exon 7 (20.9%) (Figure 2A). Regarding *EDA* variant distribution, 19 (44.2%) were in the TNF homologous domain, 10 (23.3%) were in the collagen domain, 4 (9.3%) were in the furin cleavage domain, and 3 (7.0%) were in the TM domain (Figure 2A,B).

### 2.4. Patterns of EDA-Associated Deciduous Tooth Agenesis

In our study, 97.4% (646/663) of missing deciduous teeth resulted in congenital absence of successive teeth in patients with *EDA* variants. Only 2.6% (17/663) of deciduous teeth were absent, with successive permanent teeth still developing (eight canines, five central incisors, and four lateral incisors). In patients with variants in other genes, all successive teeth were absent where deciduous tooth agenesis occurred (Appendix A).

The number of congenital absences of deciduous teeth in the four quadrants is shown in Figure 3A. The deficiency rates of the right and left sides did not significantly differ (*p* > 0.05). The number of missing deciduous teeth in the mandible was higher than that in the maxilla, with means of 8.4 ± 2.2 (mean ± SD) and 7.0 ± 2.3, respectively (*p* < 0.001). Moreover, the most frequently absent deciduous teeth associated with EDA variants were mandibular central incisors (100.0%), maxillary lateral incisors (98.8%), and mandibular lateral incisors (97.7%) (Figure 3B–D). In contrast, the maxillary second deciduous molars (40.7%) and central deciduous incisors (48.8%) had the lowest missing rates.

The mean number of missing deciduous teeth between nonsense (18.3 ± 0.6), frameshift (16.3 ± 4.3), or missense variants (15.2 ± 4.3) did not significantly differ (*p* > 0.05; Figure 4A). No significant difference (*p* > 0.05) was observed in the mean number of missing deciduous teeth between the four functional domains of EDA (Figure 4B). Concerning the missing rate of maxillary deciduous central incisors, the furin group (12.5%) had a significantly lower rate than those of the collagen (58.3%, *p* = 0.017) and TNF groups (62.0%, *p* < 0.001). Regarding the missing rate of maxillary deciduous second molars, the TM (100%) and collagen groups (83.3%) had significantly higher rates than those of the furin (31.3%, *p* = 0.026 versus TM group, *p* = 0.0093 versus collagen group) and TNF groups (32.0%, *p* = 0.030 versus TM group, *p* < 0.001 versus collagen group). Considering the missing rate of mandibular deciduous second molars, variants in the collagen domain (100%) had significantly higher rates than those in the furin (50.0%, *p* < 0.001) and TNF domains (64.0%, *p* = 0.035) (Figure 4C).

## 3. Discussion

The current understanding of the molecular regulatory mechanisms of tooth development is largely based on transgenic mouse model studies [43]. However, as monophyodonts, mice differ considerably from humans who are diphyodonts. Therefore, whether the same molecular mechanisms regulate the development of human deciduous and permanent teeth remains unknown. Notwithstanding, studies on the genotype–phenotype correlation in patients with tooth agenesis can provide clues about the similarities and differences in regulatory mechanisms underlying deciduous and permanent teeth development. Here, we identified the variants in several genes, including *EDA*, *MSX1*, *PAX9*, *LRP6*, *BMP4*, *WNT10A*, *PITX2*, and *EDARADD*, associated with deciduous tooth agenesis. *EDA* variants were present in 86.9% of deciduous tooth agenesis cases, suggesting that *EDA* is specifically required for deciduous tooth development and its variants are major pathogenic factors for deciduous tooth agenesis.

The frequency of *EDA* variants in permanent tooth agenesis is 5.9% [16], significantly lower than that in deciduous tooth agenesis (86.9%) reported in the present study, whereas *WNT10A* is the major pathogenic gene for permanent tooth agenesis, with its variants taking part in 26.0% of permanent tooth agenesis cases [16]. It is worth noting that patients with biallelic *WNT10A* loss-of-function variants showed anodontia in permanent dentition but almost normal deciduous dentition, indicating that *WNT10A* is crucial for permanent tooth development [26,44]. Our results and previous findings suggest that the development of deciduous and permanent teeth is under differential genetic regulation. Furthermore, it is noteworthy that the absence of deciduous teeth resulting from the other gene variants was relatively mild, except in the case of *EDARADD*. This observation further indicated the significance of the EDA pathway in the intricate process of deciduous tooth development. However, the different regulation mechanisms underlying deciduous and permanent tooth development need to be elucidated by further in-depth studies using transgenic diphyodont animal models, such as miniature pigs (*Sus scrofa domestica*) or ferrets (*Mustela furo*).

Deciduous tooth agenesis is more likely to occur with successive permanent tooth agenesis. Nevertheless, we found that in patients with *EDA* variants, 2.6% of successive permanent teeth developed where deciduous teeth were missing, including eight permanent canines, five permanent central incisors, and four permanent lateral incisors. Our finding is consistent with a recent study [45], demonstrating that not all deciduous teeth and their successive permanent teeth were simultaneously congenitally absent. We further analyzed the patterns of *EDA*-associated deciduous tooth agenesis. We found that the most frequently absent deciduous teeth were mandibular central incisors (100.0%), followed by maxillary lateral incisors (98.8%), mandibular lateral incisors (97.7%), and maxillary first molars (93.0%). However, the maxillary second deciduous molars (40.7%) and central deciduous incisors (48.8%) were the least likely to be missing, indicating that the requirement for EDA protein by deciduous teeth at different positions varies during development. However, given the limited sample size, a larger number of samples is necessary to substantiate these observations.

In our study, it is noteworthy that all identified *EDA* variants were observed exclusively in male patients. This observation can be attributed to the fact that XLHED and non-syndromic tooth agenesis resulting from *EDA* variants are X-linked recessive disorders [46]. These male patients’ mothers, who carried pathogenic heterozygous variants in the *EDA* gene, did not exhibit any missing teeth or only presented with mild phenotypes, such as the conical upper later incisors [47,48]. This can be explained by the presence of a second normal allele on their X chromosome, which mitigated the manifestation of these disorders.

Regarding *EDA* variant distribution in the four domains, most (44.2%) variants associated with deciduous tooth agenesis were in the TNF homologous domain, consistent with previous findings on *EDA* variant distribution associated with permanent tooth agenesis [49]. The TNF homologous domain forms a homotrimer that can bind to EDAR [21,22,23]; our findings confirmed that the TNF homologous domain is critical for EDA to exert its biological functions during tooth development.

In summary, our study, based on a large cohort, revealed that *EDA*, *PAX9*, *LRP6*, *MSX1*, *BMP4*, *WNT10A*, *PITX2*, and *EDARADD* variants are responsible for deciduous tooth agenesis. *EDA* variants accounted for 86.9% of patients with deciduous tooth agenesis. The mandibular central incisors, maxillary lateral incisors, and mandibular lateral incisors were the most frequently absent deciduous teeth associated with *EDA* variants. Our results provide valuable information on the pathogenic mechanisms underlying deciduous tooth agenesis and benefit the clinical diagnosis and genetic counseling of patients with deciduous tooth agenesis.

## 4. Materials and Methods

### 4.1. Patients

From 2001 to 2023, a total of 84 patients were recruited by the Department of Prosthodontics and the First Clinical Division of the Peking University Hospital of Stomatology. Patients included in this study presented with severe deciduous tooth agenesis and oligodontia, aged 2 to 25, including 81 males and 3 females. Oral examinations and panoramic radiographs were used to verify the number and locations of missing teeth. For patients in the deciduous dentition period, the number and pattern of missing deciduous teeth were checked and recorded by a professional dentist. For patients in the mixed and permanent dentition periods, deciduous tooth agenesis was diagnosed based on previous medical records. To reduce the potential impact of environmental factors, patients with fewer than three missing deciduous teeth were excluded. The definite environmental influencing factors influencing during pregnancy, such as radiotherapy, chemotherapy, and viral infections, were also the exclusion criteria [49,50]. Phenotypic characteristics of the scalp and body hair, skin, nails, tolerance to heat, and ability to sweat were examined. The dental condition of family members of the probands was also recorded. The dental condition of family members of the probands was also recorded. All participants or their parents provided written consent. This study was approved by the Ethics Committee of Peking University School and Hospital of Stomatology (PKUSSIRB-202162021).

### 4.2. Variant Analysis and Detection

Patients brought genetic test reports from other medical institutions to the clinic, where the disease-causing genes and variant types were recorded after detailed examination. For patients without genetic tests, genomic DNA was extracted from peripheral blood lymphocytes using a Universal Genomic DNA Kit (CWBIO, Taizhou, Jiangsu, China), as previously described [31]. Whole-exome sequencing of the samples was performed by Beijing Angen Gene Medicine Technology (Beijing, China). The detected variants were screened using the following methods. First, all genes associated with orodental disease were filtered [51]. Next, all insertions/deletions and non-synonymous single-nucleotide variants were excluded by screening out variants with a minor allele frequency ≥ 0.01 in Exome Aggregation Consortium (ExAC, http://exac.broadinstitute.org (accessed on 15 October 2023)), Single Nucleotide Polymorphism database (dbSNP, https://www.ncbi.nlm.nih.gov/snp/ (accessed on 15 October 2023)), 1000 Genomes (http://www.1000genomes.org (accessed on 15 October 2023)), and Genome Aggregation Database (gnomAD, http://gnomad.broadinstitute.org (accessed on 15 October 2023)). Subsequently, Fathmm (https://fathmm.biocompute.org.uk/inherited.html (accessed on 15 October 2023)), PolyPhen-2 (http://genetics.bwh.harvard.edu/pph2/ (accessed on 15 October 2023)), and MutationTaster (https://www.mutationtaster.org/ (accessed on 15 October 2023)) were used to further predict the pathogenicity of the variants. For variant validation and family co-segregation, the coding regions of screened genes were amplified by PCR and Sanger sequencing for probands and available family members. Primers and PCR conditions are shown in Appendix A.

### 4.3. Statistical Analysis

Because *EDA* accounted for the vast majority of variants, we conducted a separate in-depth analysis of patients in this group. The number of *EDA* variants in each exon and domain was counted separately. In 73 patients with *EDA* variants, only 43 patients had definite missing deciduous tooth positions; therefore, we compiled the number of missing teeth at different positions in four oral quadrants of 43 patients. A paired Student’s *t*-test was used to compare the missing rate in the upper and lower jaws. For different positions, comparisons of congenital deficiency rates were performed using a chi-square test. To analyze the relationship between genotype and phenotype, the mean number of missing deciduous teeth for patients with different types or domains of variants was calculated and analyzed using analysis of variance. All statistical analyses were performed using SPSS 26.0 (SPSS Inc., Chicago, IL, USA), and charts were constructed using GraphPad Prism (V8.0; GraphPad Software, La Jolla, CA, USA). Statistical significance was considered at *p* < 0.05.

## Figures and Tables

**Figure 1 ijms-25-10451-f001:**
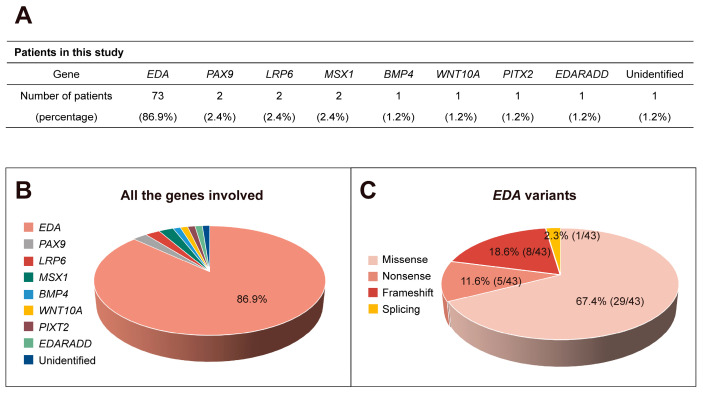
Variants found in the patient cohort. (**A**,**B**) Number and percentage of patients with different variants identified in this study. (**C**) Proportion of different *EDA* variants.

**Figure 2 ijms-25-10451-f002:**
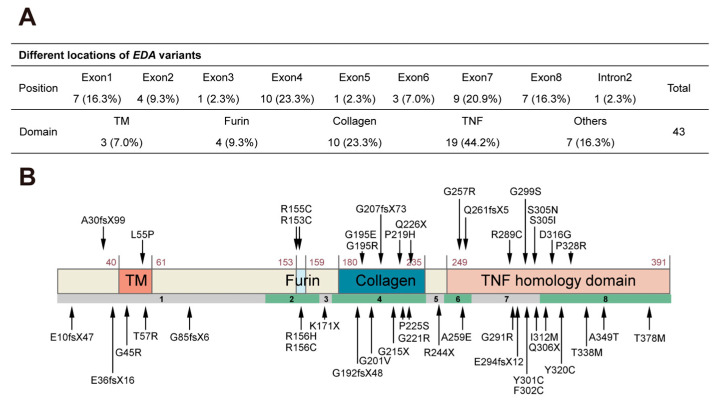
Localization of the 43 *EDA* variants. (**A**) Number and distribution of *EDA* variants in different exons and domains. (**B**) Schematic diagram of the wild-type ectodysplasin A protein and the localization of the *EDA* variants identified in this study. TM, transmembrane; TNF, tumor necrosis factor.

**Figure 3 ijms-25-10451-f003:**
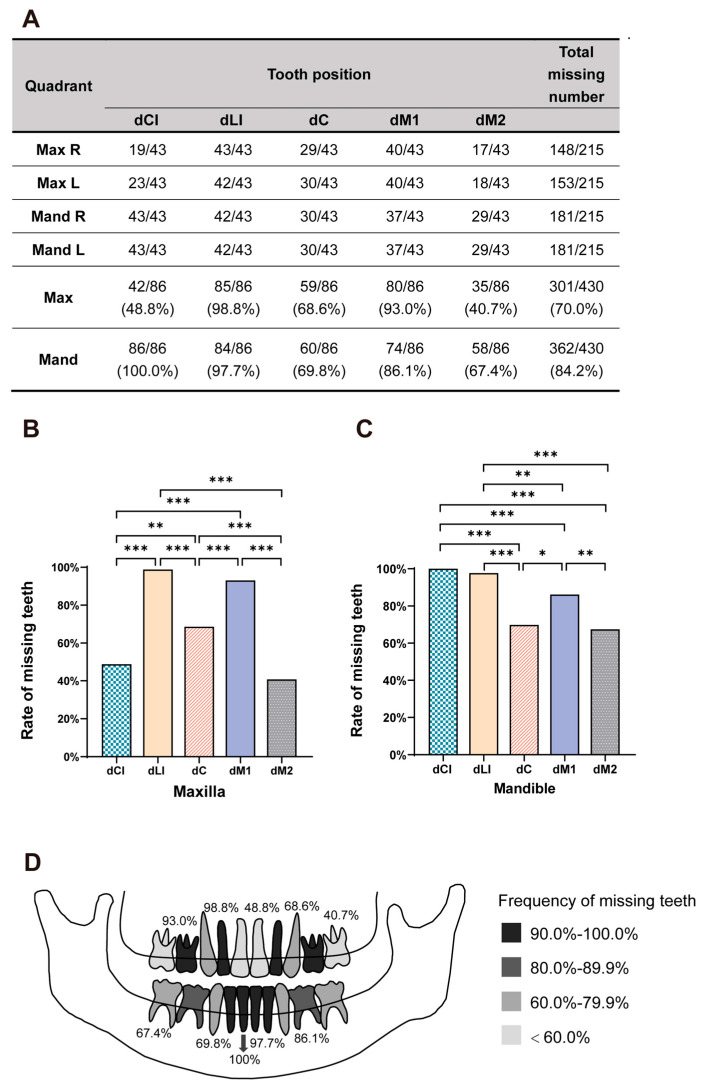
Dental phenotype of the 43 patients with *EDA* variants. (**A**) Count and percentage of missing deciduous teeth at each position in maxillary and mandibular dentition. (**B**,**C**) Proportion of absence at five positions in maxillary and mandibular dentition, respectively. Asterisks indicate significant differences (* *p* < 0.05, ** *p* < 0.01, *** *p* < 0.001). (**D**) Schema showing the missing frequencies at each deciduous tooth position. A darker color indicates higher rates. Max, maxillary; Mand, mandibular; R, right; L, left; dCI, deciduous central incisor; dLI, deciduous lateral incisor; dC, deciduous canine; dM1, first deciduous molar; dM2, second deciduous molar.

**Figure 4 ijms-25-10451-f004:**
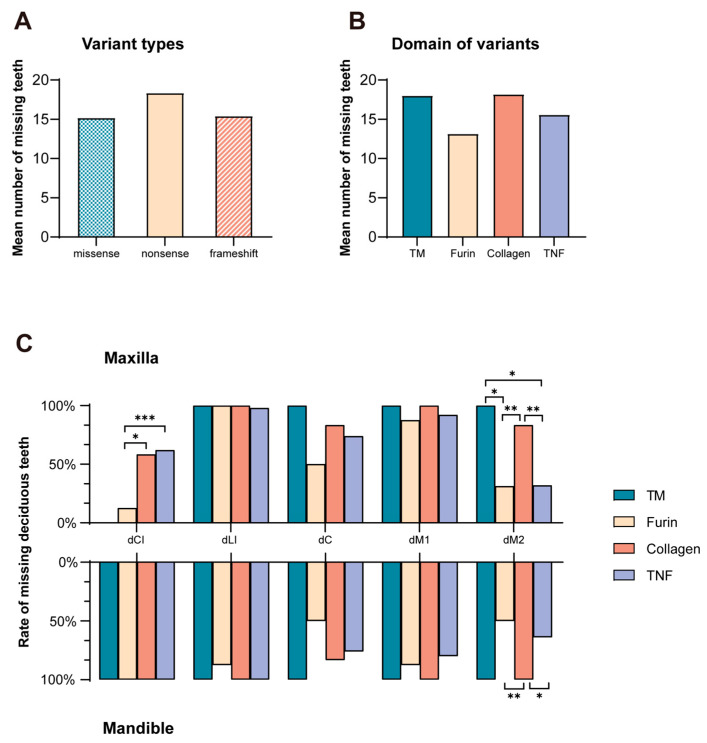
Distribution of missing deciduous teeth due to different *EDA* variant types and locations. (**A**) Mean number of missing deciduous teeth in patients with different *EDA* variant types. (**B**) Mean number of missing deciduous teeth in patients with *EDA* variants situated at the four domains. (**C**) Rates of maxillary and mandibular missing deciduous teeth at five positions caused by variants in different domains. Asterisks indicate significant differences (* *p* < 0.05, ** *p* < 0.01, *** *p* < 0.001). TM, transmembrane domain; TNF, tumor necrosis factor homologous domain; dCI, deciduous central incisor; dLI, deciduous lateral incisor; dC, deciduous canine; dM1, first deciduous molar; dM2, second deciduous molar.

**Table 1 ijms-25-10451-t001:** Variants and phenotypes of patients in this study.

Patient	Gender/Age	Gene	Exon	Nucleotide Change	Protein Change	Variant Type	Domain	Syndromic	Family History	Number of Malformed Teeth in the Deciduous Dentition	Number of Missing Teeth in the Deciduous Dentition	Number of Missing Teeth in the Permanent Dentition (Excluding Third Molar)
1	M, 5	** *EDA* **	1	c.170C > G	p.T57R	missense	TM	XLHED	+	2	18	26
2	M, 4	** *EDA* **	8	c.1133C > T	p.T378M	missense	TNF	XLHED	+	2	16	21
3	M, 12	** *EDA* **	8	c.959A > G	p.Y320C	missense	TNF	XLHED	+	0	16	18
4	M, 15	** *EDA* **	8	c.1133C > T	p.T378M	missense	TNF	XLHED	-	unk	unk	26
5	M, 4	** *EDA* **	7	c.882_885del	p.E294Dfs*12	frameshift	TNF	XLHED	+	0	20	28
6	M, 15	** *EDA* **	7	c.902A > G	p.Y301C	missense	TNF	XLHED	?	unk	unk	24
7	M, 14	** *EDA* **	1	c.252del	p.G85Afs*6	frameshift		XLHED	+	unk	unk	21
8	M, 6	** *EDA* **	8	c.947A > G	p.D316G	missense	TNF	XLHED	+	0	20	28
9	M, 22	** *EDA* **	2	c.467G > A	p.R156H	missense	Furin	XLHED	+	unk	unk	20
10	M, 12	** *EDA* **	2	c.467G > A	p.R156H	missense	Furin	XLHED	+	unk	unk	21
11	M, 9	** *EDA* **	8	c.1013C > T	p.T338M	missense	TNF	NSTA	+	unk	unk	12
12	M, 6	** *EDA* **	8	c.1133C > T	p.T378M	missense	TNF	XLHED	+	4	14	24
13	M, 10	** *EDA* **	6	c.776C > A	p.A259E	missense	TNF	NSTA	?	unk	unk	15
14	M, 3	** *EDA* **	8	c.1045G > A	p.A349T	missense	TNF	XLHED	+	2	16	19
15	M, 3	** *EDA* **	4	c.643G > T	p.G215*	nonsense	Collagen	XLHED	+	1	19	26
16	M, 12	** *EDA* **	2	c.457C > T	p.R153C	missense	Furin	XLHED	+	unk	unk	13
17	M, 4	** *EDA* **	1	c.106_118del	p.E36Afs*16	frameshift		XLHED	-	4	12	28
18	M, 3	** *EDA* **	6	c.769G > C	p.G257R	missense	TNF	NSTA	+	0	14	22
19	M, 3	** *EDA* **	6	c.769G > C	p.G257R	missense	TNF	NSTA	+	0	14	22
20	M, 11	** *EDA* **	7	c.914G > T	p.S305I	missense	TNF	XLHED	+	0	20	28
21	M, 22	** *EDA* **	4	c.602G > T	p.G201V	missense	Collagen	XLHED	?	unk	unk	22
22	M, 4	** *EDA* **	7	c.865C > T	p.R289C	missense	TNF	XLHED	+	4	9	22
23	M, 8	** *EDA* **	8	c.1045G > A	p.A349T	missense	TNF	XLHED	-	unk	unk	28
24	M, 14	** *EDA* **	7	c.895G > A	p.G299S	missense	TNF	XLHED	?	unk	unk	26
25	M, 11	** *EDA* **	7	c.895G > A	p.G299S	missense	TNF	XLHED	?	unk	unk	20
26	M, 20	** *EDA* **	8	c.1133C > T	p.T378M	missense	TNF	XLHED	?	unk	unk	26
27	M, 20	** *EDA* **	8	c.1133C > T	p.T378M	missense	TNF	XLHED	+	unk	unk	25
28	M, 3	** *EDA* **	7	c.871G > A	p.G291R	missense	TNF	XLHED	?	0	20	28
29	M, 7	** *EDA* **	8	c.936C > G	p.I312M	missense	TNF	NSTA	+	unk	unk	17
30	M, 18	** *EDA* **	1	c.133G > C	p.G45R	missense	TM	NSTA	?	unk	unk	15
31	M, 7	** *EDA* **	1	c.88_89insG	p.A30Gfs*99	frameshift		XLHED	+	unk	unk	26
32	M, 16	** *EDA* **	1	c.28del ^#^	p.E10Nfs*47	frameshift		XLHED	-	unk	unk	27
33	M, 4	** *EDA* **	7	c.916C > T	p.Q306*	nonsense	TNF	XLHED	+	0	18	28
34	M, 7	** *EDA* **	2	c.463C > T	p.R155C	missense	Furin	XLHED	+	unk	unk	18
35	M, 4	** *EDA* **	7	c.895G > A	p.G299S	missense	TNF	XLHED	+	0	20	28
36	M, 11	** *EDA* **	8	c.1013C > T	p.T338M	missense	TNF	NSTA	+	unk	unk	12
37	M, 9	** *EDA* **	4 4	c.656C > A ^#^ c.661G > C ^#^	p.P219H p.G221R	missense missense	Collagen Collagen	XLHED	-	unk	unk	27
38	M, 18	** *EDA* **	7	c.914G > A ^#^	p.S305N	missense	TNF	XLHED	+	0	18	23
39	M, 16	** *EDA* **	2	c.463C > T	p.R155C	missense	Furin	XLHED	+	unk	unk	19
40	M, 5	** *EDA* **	2	c.463C > T	p.R155C	missense	Furin	XLHED	+	6	10	16
41	M, 5	** *EDA* **	2	c.457C > T	p.R153C	missense	Furin	XLHED	?	8	4	17
42	M, 6	** *EDA* **	4 4	c.673C > T c.676C > T	p.P225S P.Q226*	missense nonsense	Collagen Collagen	XLHED	+	2	18	25
43	M, 21	** *EDA* **	3	c.511A > T ^#^	p.K171*	nonsense		XLHED	?	unk	unk	19
44	M, 25	** *EDA* **	5	c.730C > T	p.R244*	nonsense		XLHED	+	unk	unk	28
45	M, 2	** *EDA* **	8	c.1133C > T	p.T378M	missense	TNF	XLHED	+	0	20	25
46	M, 5	** *EDA* **	8	c.1133C > T	p.T378M	missense	TNF	XLHED	+	4	14	23
47	M, 4	** *EDA* **	8	c.1133C > T	p.T378M	missense	TNF	XLHED	-	0	20	28
48	M, 3	** *EDA* **	8	c.983C > G ^#^	p.P328R	missense	TNF	XLHED	+	0	10	17
49	M, 6	** *EDA* **	2	c.457C > T	p.R153C	missense	Furin	XLHED	+	4	10	16
50	M, 6	** *EDA* **	7	c.905T > G	p.F302C	missense	TNF	XLHED	+	2	10	13
51	M, 6	** *EDA* **	2	c.466C > T	p.R156C	missense	Furin	XLHED	+	0	19	27
52	M, 4	** *EDA* **	8	c.1013C > T	p.T338M	missense	TNF	NSTA	+	0	14	22
53	M, 6	** *EDA* **	1	c.164T > C	p.L55P	missense	TM	XLHED	+	2	18	26
54	M, 22	** *EDA* **	1	c.164T > C	p.L55P	missense	TM	XLHED	+	unk	unk	25
55	M, 8	** *EDA* **	4	c.619del	p.G207Efs*73	frameshift	Collagen	XLHED	+	4	13	21
56	M, 16	** *EDA* **	2	c.457C > T	p.R153C	missense	Furin	XLHED	+	unk	unk	16
57	M, 8	** *EDA* **	2	c.463C > T	p.R155C	missense	Furin	XLHED	+	unk	unk	16
58	M, 6	** *EDA* **	4	c.583G > A ^#^	p.G195R	missense	Collagen	XLHED	+	0	20	28
59	M, 6	** *EDA* **	7	c.871G > A	p.G291R	missense	TNF	XLHED	+	0	20	26
60	M, 4	** *EDA* **	8	c.1045G > A	p.A349T	missense	TNF	XLHED	+	1	17	28
61	M, 4	** *EDA* **	2	c.467G > A	p.R156H	missense	Furin	XLHED	-	2	16	22
62	M, 5	** *EDA* **	4	c.572dup ^#^	p.G192Rfs*48	frameshift	Collagen	XLHED	+	0	20	28
63	M, 3	** *EDA* **	4	c.584G > A	p.G195E	missense	Collagen	XLHED	+	1	19	25
64	M, 12	** *EDA* **	2	c.457C > T	p.R153C	missense	Furin	XLHED	?	unk	unk	13
65	M, 23	** *EDA* **	2	c.467G > A	p.R156H	missense	Furin	XLHED	-	0	18	23
66	M, 5	** *EDA* **	2	c.467G > A	p.R156H	missense	Furin	XLHED	+	6	14	24
67	M, 3	** *EDA* **	6	c.781dup ^#^	p.Q261Pfs*5	frameshift	TNF	XLHED	+	6	12	25
68	M, 8	** *EDA* **	8	c.1013C > T	p.T338M	missense	TNF	NSTA	+	0	8	17
69	M, 4	** *EDA* **	2	c.463C > T	p.R155C	missense	Furin	XLHED	+	2	14	22
70	M, 4	** *EDA* **		c.502 + 1G > A				XLHED	+	4	12	24
71	M, 23	** *EDA* **	2	c.467G > A	p.R156H	missense	Furin	XLHED	?	unk	unk	18
72	M, 8	** *EDA* **	3	c.511A > T ^#^	p.K171*	nonsense		XLHED	-	unk	unk	24
73	M, 12	** *EDA* **	8	c.1013C > T	p.T338M	missense	TNF	NSTA	?	0	9	17
74	F, 22	** *P* ** ** *A* ** ** *X9* **	2	c.236_237insAC	p.T80Lfs*6	frameshift	Paired DNA-binding	NSTA	+	unk	4	16
75	M, 18	** *P* ** ** *A* ** ** *X9* **	2	c.336C > A ^#^	p.C112*	nonsense		NSTA	+	unk	unk	15
76	M, 11	** *LRP6* **	11	c.2292G > A	p.W764*	nonsense	β-propeller	HED	+	0	6	18
77	M, 18	** *LRP6* **	4	c.716G > A ^#^	p.W239*	nonsense	β-propeller	NSTA	+	0	4	6
78	M, 12	** *MSX1* **	2	c.670C > T	p.R224C	missense	HD	NSTA	+	0	2	17
79	M, 12	** *MSX1* **	1	c.421del ^#^	p.E141Rfs*19	frameshift		NSTA	?	0	4	15
80	M, 9	** *BMP4* **	4	c.614T > C	p.V205A	missense	TGFβ propeptide	eye anomaly	+	unk	unk	16
81	F, 9	** *WNT10A* **	4	c.826T > A c.949del	p.C276S p.A317Hfs*121	missense frameshift		OODD	+	4	2	28
82	F, 5	** *PITX2* **	3	c.630insCG	p.V211Rfs*28	frameshift	TAD2	ARS	-	0	9	22
83	M, 4	** *EDARADD* **	4	c.208_209ins AGAATAATTT ^#^	p.M70Kfs*5	frameshift		HED	+	2	18	26
84	M, 5	**Undefined**						NSTA	-	0	20	19

M, male; F, female; TM, transmembrane domain; TNF, tumor necrosis factor homologous domain; HD, homeodomain; TAD2, transcriptional activation domain 2 domain; XLHED, X-linked hypohidrotic ectodermal dysplasia; NSTA, non-syndromic tooth agenesis; HED, hypohidrotic ectodermal dysplasia; OODD, odonto-onycho-dermal dysplasia; ARS, Axenfeld–Rieger syndrome; unk, unknown; pound (#) keys mark the novel variants; question marks (?) indicate uncertainty.

## Data Availability

The original contributions presented in the study are included in the Appendix A; further inquiries can be directed to the corresponding authors.

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
