# Peer review of "EDA Variants Are Responsible for Approximately 90% of Deciduous Tooth Agenesis"

_ijms, 2024, doi:10.3390/ijms251910451_

Round 1
Reviewer 1 Report
Comments and Suggestions for Authors
Thank you for inviting me to review the manuscript. I hope to collaborate on some considerations, questions and suggestions.
The study aimed to identify the pathogenic genes and clinical features of patients with deciduous tooth agenesis. Overall, the article is exciting and addresses a topic that is still not very clear.
The introduction states: “ Moreover, patients with severe deciduous tooth agenesis are often associated with systemic symptoms, as well as more serious permanent tooth agenesis, resulting in severe alveolar bone atrophy and maxillofacial bone underdevelopment[3]”– Question: How was the general clinical evaluation for the diagnosis of possible syndromes in which dental agenesis is part of the clinical phenotype?
The methodology stated: “To reduce the potential impact of environmental factors, patients with less than three missing deciduous teeth were excluded.” Question: What environmental factor could lead to dental agenesis in deciduous dentition? In my opinion, depending on the child's age at the first dental appointment, and especially if there was follow-up, even with only one missing tooth, the child could be included in the study, and this is a suggestion for future studies. Therefore, considering that the study has already been carried out, it is necessary to explain in the text what this environmental factor (considered in the exclusion criteria).
Another aspect of weakness in the article, still about the methodology, concerns the inclusion of children in the mixed or permanent dentition phase since the records in the medical records often do not present all the information necessary for the accurate diagnosis of dental agenesis and it is not possible to confirm dental agenesis in the deciduous dentition.
In the Results, regardless of Table 1, since patients with ectodermal dysplasia and patients without dysplasia were included, one suggestion would be to separate the analyses of the variants, separating the group into patients with ectodermal dysplasia and the group of patients with non-syndromic agenesis, presenting the results for each group separately. Another suggestion would be for the cases of ectodermal dysplasia to be evaluated by a geneticist to define possible syndromes with these phenotypes in the characterization. It was also difficult to see in Table 1 which gene corresponds to each mutation. I believe that the file may have been misconfigured. It is essential to review this.
The methodology mentioned in line 262: “For variant validation and family co-segregation...” the question: Were family members of probands with dental agenesis evaluated and recruited? If this was the approach, this information should be detailed in the methodology.
The statistical analysis states in line 268: “For 43 out of 73 patients with EDA variants, we compiled the number of missing teeth at different positions in four oral quadrants.” Question: Why were missing teeth characterizations compiled for only 43 patients? Why not characterize all 73 who presented EDA variants?
The discussion presents several data and comparisons on the findings of variants in the EDA gene. However, although fewer variants were found in other genes, they could also have been briefly discussed.
An important aspect that was also not discussed was that numerous variants were found in the EDA gene, which, as mentioned in the introduction, is located on the X chromosome. Thus, the fact that this gene is located on the X chromosome is related to the fact that these variants, or most of them, were found in male patients, in whom, when the family was investigated, the mothers had agenesis, typical of phenotypes related to genes present on the X chromosome. However, this aspect was not addressed in the discussion; I suggest discussing this point.
About the supplementary material
In Table S1, it is necessary to include a caption showing what * and # mean. Likewise, the purpose of including a reference in this table is not clear. It is very confusing, did these cited authors publish studies finding these same variants? Or were these findings from these authors, which have already been published, included as if they were from this study? If the second option is the case, the article being evaluated for the International Journal of Molecular Science should only include the cases analysed in this study, removing from the case series all the findings from other authors in previously published Papers.
Thus, analyzing Table S1 in detail, it appears that the present study generated data at this time for 43 cases (cited as this study), and of these, only 29 of them had information about the deciduous dentition, and the other 14 in the deciduous dentition appear “cannot defined”. Evaluating from this point of view, so that the article is in line with the title, I believe that the article should be completely revised, presenting, analyzing and discussing the data from only these 29 cases. The article is interesting and I believe that with this major revision it could be aligned with the title and the objective of the work. However, if the option is to keep it as it is, the title and objective need to be reformulated, because in this context it would not be evaluating only the deciduous dentition and would be a mix of research article and literature review.
Author Response
Comments 1: The introduction states: “ Moreover, patients with severe deciduous tooth agenesis are often associated with systemic symptoms, as well as more serious permanent tooth agenesis, resulting in severe alveolar bone atrophy and maxillofacial bone underdevelopment[3]”– Question: How was the general clinical evaluation for the diagnosis of possible syndromes in which dental agenesis is part of the clinical phenotype?
Response 1: Thank you for pointing this out. In addition to the tooth agenesis, we also recorded the developmental abnormalities of other tissues in the patients, encompassing but not limited to hair, sweat glands, lacrimal glands, saliva, iris, nails, skin and other systemic defects. For hypohidrotic ectodermal dysplasia (HED), it is characterized by a triad of typical signs comprising sparse hair (hypotrichosis), abnormal or missing teeth (anodontia or hypodontia), and inability to sweat (anhidrosis or hypohidrosis).
Comments 2: The methodology stated: “To reduce the potential impact of environmental factors, patients with less than three missing deciduous teeth were excluded.” Question: What environmental factor could lead to dental agenesis in deciduous dentition? In my opinion, depending on the child's age at the first dental appointment, and especially if there was follow-up, even with only one missing tooth, the child could be included in the study, and this is a suggestion for future studies. Therefore, considering that the study has already been carried out, it is necessary to explain in the text what this environmental factor (considered in the exclusion criteria).
Response 2: Thank you for valuable advice. We agree with your view, as a recent study shown that in the majority of cases, only one or two deciduous teeth were missing because of the influence of environmental factors. Notably, existing literature suggests that environmental factors such as prenatal exposure to radiotherapy, chemotherapy, and viral infections can influence the occurrence of missing primary teeth. To facilitate genetic analysis and study the effect of gene variants on teeth agenesis, we have incorporated these exclusion criteria into the Materials and Methods section 4.1.
Comments 3: Another aspect of weakness in the article, still about the methodology, concerns the inclusion of children in the mixed or permanent dentition phase since the records in the medical records often do not present all the information necessary for the accurate diagnosis of dental agenesis and it is not possible to confirm dental agenesis in the deciduous dentition.
In the Results, regardless of Table 1, since patients with ectodermal dysplasia and patients without dysplasia were included, one suggestion would be to separate the analyses of the variants, separating the group into patients with ectodermal dysplasia and the group of patients with non-syndromic agenesis, presenting the results for each group separately. Another suggestion would be for the cases of ectodermal dysplasia to be evaluated by a geneticist to define possible syndromes with these phenotypes in the characterization. It was also difficult to see in Table 1 which gene corresponds to each mutation. I believe that the file may have been misconfigured. It is essential to review this.
Response 3: Thank you for providing such detailed advice. In our study, we verified the congenital absence of deciduous teeth in children during the mixed or permanent dentition phase by previous medical records of deciduous dentition from other hospital. And some patients have serious tooth agenesis, such as anodontia. The patients we could not confirm the tooth position of dental agenesis (30/73), which marked “unknown” in table 1, were not included in the analysis of pattern of deciduous tooth agenesis.
To enhance clarity, we have added a Supplementary Figure to differentiate between patients with syndromic and non-syndromic tooth agenesis, and relevant findings have been incorporated into Result 2.1. Moreover, in Result 2.2, we have extensively discussed the phenotypes of patients with ectodermal dysplasia or non-syndromic tooth agenesis.
We wholeheartedly agree with your suggestion regarding the involvement of geneticists in diagnosing these cases. Our team has been dedicated to the study of tooth agenesis for over two decades, and the cases involving ectodermal dysplasia have been evaluated by multiple dentists. Additionally, we have sought consultations with geneticists, dermatologists, ophthalmologists, and other specialists, although whose names are not included in the author list. Further details regarding the specific phenotypes can be found in Table S2.
Taking your feedback into account, we have revised the format of Table 1 to provide clearer correspondence with different genes.
Comments 4: The methodology mentioned in line 262: “For variant validation and family co-segregation...” the question: Were family members of probands with dental agenesis evaluated and recruited? If this was the approach, this information should be detailed in the methodology.
Response 4: Thanks for your detailed advice. In our study, the phenotypes of certain family members of probands were recorded, and PCR and Sanger sequencing techniques were employed to amplify the respective genes. Considering the length of the article, we opted to present their results succinctly under the "family history" section in Table 1.
As your suggested, we have added more information in Materials and Methods sections 4.1 and 4.2.
Comments 5: The statistical analysis states in line 268: “For 43 out of 73 patients with EDA variants, we compiled the number of missing teeth at different positions in four oral quadrants.” Question: Why were missing teeth characterizations compiled for only 43 patients? Why not characterize all 73 who presented EDA variants?
Response 5: Thanks for your advice. For children in the mixed or permanent dentition phase who were lack of detailed information during the primary dentition period, we could only infer the presence of a considerable number of missing primary teeth based on their intraoral conditions and the limited available information. We were unable to precisely determine the specific locations of the missing primary teeth. Therefore, we only analyzed 43 patients for missing teeth characterizations.
As your suggested, we have added detailed information in Materials and Methods sections 4.3.
Comments 6: The discussion presents several data and comparisons on the findings of variants in the EDA gene. However, although fewer variants were found in other genes, they could also have been briefly discussed.
Response 6: Thanks for pointing this out. Therefore, we have added the corresponding parts of discussion in lines 201-204. According to our findings, the absence of deciduous teeth resulting from the other gene variants was relatively mild, with the exception of EDARADD. This indicated that the EDA pathway may play an important role in the development of deciduous teeth. However, additional data is indispensable for a more comprehensive analysis of this phenomenon.
Comments 7: An important aspect that was also not discussed was that numerous variants were found in the EDA gene, which, as mentioned in the introduction, is located on the X chromosome. Thus, the fact that this gene is located on the X chromosome is related to the fact that these variants, or most of them, were found in male patients, in whom, when the family was investigated, the mothers had agenesis, typical of phenotypes related to genes present on the X chromosome. However, this aspect was not addressed in the discussion; I suggest discussing this point.
Response 7: Thanks for your valuable advice. Considering that XLHED and non-syndromic tooth agenesis caused by EDA variants are X-linked recessive disorders, the majority of females with pathogenic heterozygous variants in EDA tend to exhibit either no missing teeth or mild phenotypes. We have added the corresponding parts of discussion in lines 222-228.
Comments 8: About the supplementary material
In Table S1, it is necessary to include a caption showing what * and # mean. Likewise, the purpose of including a reference in this table is not clear. It is very confusing, did these cited authors publish studies finding these same variants? Or were these findings from these authors, which have already been published, included as if they were from this study? If the second option is the case, the article being evaluated for the International Journal of Molecular Science should only include the cases analysed in this study, removing from the case series all the findings from other authors in previously published Papers.
Thus, analyzing Table S1 in detail, it appears that the present study generated data at this time for 43 cases (cited as this study), and of these, only 29 of them had information about the deciduous dentition, and the other 14 in the deciduous dentition appear “cannot defined”. Evaluating from this point of view, so that the article is in line with the title, I believe that the article should be completely revised, presenting, analyzing and discussing the data from only these 29 cases. The article is interesting and I believe that with this major revision it could be aligned with the title and the objective of the work. However, if the option is to keep it as it is, the title and objective need to be reformulated, because in this context it would not be evaluating only the deciduous dentition and would be a mix of research article and literature review.
Response 8: Thank you for your careful examinations. We have appended the symbol definitions at the conclusion of the table.
In our research, all the patients have been recruited by our team in the Department of Prosthodontics and the First Clinical Division of the Peking University Hospital of Stomatology. Some patients we collected have been previously published by our team. In order to conduct this study with a substantial sample size, we have consolidated the data of all patients from 2001 onwards. Corresponding clarifications have been integrated at the termination of Table S1.
Reviewer 2 Report
Comments and Suggestions for Authors
check the attached word document

Author Response
Comments 1: I recommend changing the title to more attractive as well as presentable one. The present title is a statement not a title for a manuscript. please add the type of the study ''prospective cohort''.
Response 1: Thanks for your valuable advice. Actually, our study was a retrospective study rather than a prospective cohort study, as it focused on past data rather than following participants into the future. It is clear that the variants and tooth agenesis were already present at the time of the patients' initial visits. Over the course of the past two decades, we have continuously collected data on these patients.
Comments 2: The total period of the study is not clear.
Response 2: Thanks for pointing this out. Since 2001, we have consistently collected patients with congenital absence of deciduous teeth. These patients sought treatment at the Department of Prosthodontics in the Peking University Hospital of Stomatology and were enrolled in our study. Oral examination and panoramic dental radiographs were performed and peripheral blood samples was collected with the patient's informed consent.
Comments 3: What was the age range for patient recruitment?
Response 3: Thank you for providing further details. The age range of our patients in this study varied from 2 to 25 years old. The age of the patients was determined based on their initial dental visit to our hospital. It is important to note that these patients sought treatment at our hospital independently, rather than being specifically recruited within a certain age range.
Comments 4: Did the authors follow all the patients till the age of 25 years?
Response 4: Thank you for your inquiry. Our study is retrospective in nature and does not mandate follow-ups until the age of 25. Nevertheless, a significant portion of our patients regularly visit our hospital for oral examinations and treatment. For individuals with congenital tooth agenesis, this treatment necessitates lifelong maintenance and care.
Comments 5: ''Patients included in this study presented with severe deciduous tooth agenesis and oligo- 234 dontia.'' The authors said. How the authors did diagnosed teeth agenesis at the age of 1 year old? Did they followed the patients up to
Response 5: Thank you for highlighting this aspect. In our research study, the youngest patient was 2 years old. Typically, all primary teeth should erupt by the age of two and a half years. Additionally, panoramic X-ray imaging was employed to further confirm both the deciduous teeth and the developing permanent tooth germs. The panoramic X-rays were retaken in every annual follow-up visits to monitor any changes or developments.
Comment 6:In table 1, the number of missed teeth was represented either by a definite number or ''unknown''? How it can be unknown?
Response 6: Thanks for your inquiry. In our study, some children in the mixed or permanent dentition phase who were lack of detailed information during the primary dentition period, we could only define that there were missing primary teeth based on their intraoral conditions and the limited available information. We were unable to precisely determine the specific locations of the missing teeth, and marked them “unknown” in table 1.
Comments 7: Did the authors recorded only if the teeth were present or not? Or they targets if its growth and complexion of its structure? The examined patients were at the age range1-8 years in which the permanent teeth formation and eruption could not be completed. Only one patient was at the age of 11 years.
Response 7: As Table 1 shown, we also recorded the number of malformed teeth in the deciduous dentition. We determined the deciduous tooth agenesis and oligodontia in permanent dentitions not only by the oral examination, but also, more importantly, by the panoramic X-ray examination. For the patients ageing below 7, although the permanent teeth may not erupt in the oral cavity, the number of the developing permanent tooth germs can be identified in panoramic X-ray image.
Reviewer 3 Report
Comments and Suggestions for Authors
Dear authors,
congratulations for the interesting chosen topic - the temporary teeth agenesis has a particular impact on the psycho-somatic development of children.
Do you consider that the female gender is exempted/has a lower incidence of this anomaly?
Is the genetic analysis recommended to be performed at a certain age, in order to allow patient monitoring?
Is there the possibility of correlations between the genetic specificity of the parents and the mode of expression of EDA?
Author Response
Comments 1: Do you consider that the female gender is exempted/has a lower incidence of this anomaly?
Response 1: Thanks for your question. According to our results, EDA variants account for approximately 90% of deciduous tooth agenesis with oligodontia. Given that EDA gene is located on the X chromosome, its impact is mainly in males, increasing the likelihood of severe tooth agenesis in male patients. This finding was consistent with the trends we observed in the clinic.
Comments 2: Is the genetic analysis recommended to be performed at a certain age, in order to allow patient monitoring?
Response 2: Thank you for pointing this out, it’s a special view. We have taken a proactive step by integrating genetic counseling into our clinical practice. Specifically, for patients who have a familial background of oligodontia or related syndromes, we will recommend genetic testing before having a child to reduce the risk of hereditary conditions. It is worth noting that because the EDA gene is situated on the X chromosome and manifests as lateral incisor malformations in females, we also place emphasis on the importance of patients' familial history in our discussions with them.
Comments 3: Is there the possibility of correlations between the genetic specificity of the parents and the mode of expression of EDA?
Response 3: Thank you for bringing up this question. It is indeed a fascinating area of research that we have thoroughly explored. At present, we have found that if parents carry genetic variants that significantly impair the function of the EDA protein, their child will exhibit a severe phenotype. Conversely, if the parents carry variants that have a milder impact on protein function, the child's phenotype will be correspondingly milder. It is important to note that dental development is a complex regulatory process involving multiple genes, and the effect of these genes is cumulative. For example, in the case of patient #81 in our study, the presence of compound heterozygous variants led to a more severe phenotype.
Moreover, aside from genetic factors, it is possible that epigenetic marks in the parents might contribute to distinct patterns of EDA gene expression and influence the phenotype of the child. We are committed to further investigating these interesting aspects in future studies.